# Vaginal Microbiome Metagenome Inference Accuracy: Differential Measurement Error according to Community Composition

Kayla A. Carter,[a] Anthony A. Fodor,[b] Jennifer E. Balkus,[a,c] Angela Zhang,[d] Myrna G. Serrano,[e,f] Gregory A. Buck,[e,f,g] Stephanie M. Engel,[h] Michael C. Wu,[i] Shan Sun[b]

[a]Department of Epidemiology, University of Washington, Seattle, Washington, USA

[b]Department of Bioinformatics and Genomics, University of North Carolina at Charlotte, Charlotte, North Carolina, USA

[c]Vaccine and Infectious Disease Division, Fred Hutchinson Cancer Research Center, Seattle, Washington, USA

[d]Department of Biostatistics, University of Washington, Seattle, Washington, USA

[e]Department of Microbiology and Immunology, School of Medicine, Virginia Commonwealth University, Richmond, Virginia, USA

[f]Center for Microbiome Engineering and Data Analysis, Virginia Commonwealth University, Richmond, Virginia, USA

[g]Department of Computer Science, College of Engineering, Virginia Commonwealth University, Richmond, Virginia, USA

[h]Department of Epidemiology, University of North Carolina at Chapel Hill, Chapel Hill, North Carolina, USA

[i]Public Health Sciences Division, Fred Hutchinson Cancer Research Center, Seattle, Washington, USA

**ABSTRACT** Several studies have compared metagenome inference performance in different human body sites; however, none specifically reported on the vaginal microbiome. Findings from other body sites cannot easily be generalized to the vaginal microbiome due to unique features of vaginal microbial ecology, and investigators seeking to use metagenome inference in vaginal microbiome research are "flying blind" with respect to potential bias these methods may introduce into analyses. We compared the performance of PICRUSt2 and Tax4Fun2 using paired 16S rRNA gene amplicon sequencing and whole-metagenome sequencing data from vaginal samples from 72 pregnant individuals enrolled in the Pregnancy, Infection, and Nutrition (PIN) cohort. Participants were selected from those with known birth outcomes and adequate 16S rRNA gene amplicon sequencing data in a case-control design. Cases experienced early preterm birth (<32 weeks of gestation), and controls experienced term birth (37 to 41 weeks of gestation). PICRUSt2 and Tax4Fun2 performed modestly overall (median Spearman correlation coefficients between observed and predicted KEGG ortholog [KO] relative abundances of 0.20 and 0.22, respectively). Both methods performed best among *Lactobacillus crispatus*-dominated vaginal microbiotas (median Spearman correlation coefficients of 0.24 and 0.25, respectively) and worst among *Lactobacillus iners*-dominated microbiotas (median Spearman correlation coefficients of 0.06 and 0.11, respectively). The same pattern was observed when evaluating correlations between univariable hypothesis test $P$ values generated with observed and predicted metagenome data. Differential metagenome inference performance across vaginal microbiota community types can be considered differential measurement error, which often causes differential misclassification. As such, metagenome inference will introduce hard-to-predict bias (toward or away from the null) in vaginal microbiome research.

**IMPORTANCE** Compared to taxonomic composition, the functional potential within a bacterial community is more relevant to establishing mechanistic understandings and causal relationships between the microbiome and health outcomes. Metagenome inference attempts to bridge the gap between 16S rRNA gene amplicon sequencing and whole-metagenome sequencing by predicting a microbiome's gene content based on its taxonomic composition and annotated genome sequences of its members. Metagenome inference methods have been evaluated primarily among gut samples, where they appear to perform fairly well. Here, we show that metagenome inference

Address correspondence to Kayla A. Carter, kaycart@uw.edu.

The authors declare a conflict of interest. J.E.B. has received honorarium from Ferring Pharmaceuticals. G.A.B. is on the Scientific Advisory Board of Juno Bio, Ltd. All other authors declare that they do not have any conflicts of interest.

performance is markedly worse for the vaginal microbiome and that performance varies across common vaginal microbiome community types. Because these community types are associated with sexual and reproductive outcomes, differential metagenome inference performance will bias vaginal microbiome studies, obscuring relationships of interest. Results from such studies should be interpreted with substantial caution and the understanding that they may over- or underestimate associations with metagenome content.

**KEYWORDS** *Lactobacillus crispatus*, *Lactobacillus iners*, measurement error, metagenome inference, vaginal microbiome

Several studies have compared the performance of various metagenome inference methods in different contexts, including human body sites and environmental ecosystems (1–7). These studies consistently report that performance improves with increasing representation of taxa in reference sequence databases (16S rRNA gene and whole genome), which results in higher-quality metagenome inference for more-well-studied ecosystems (1–7). Metagenome inference methods tend to perform best for human gut and oral microbiomes and less well for other human body sites and other mammalian gut and environmental microbiomes (1–7). However, no prior studies specifically reported on metagenome inference for the vaginal microbiome (although one reported on urogenital samples but gave no additional details on these samples) (4).

Findings from other body sites cannot easily be generalized to the vaginal microbiome due to unique features of vaginal microbial ecology. For many individuals of reproductive age, the vaginal microbiome is dominated by a single *Lactobacillus* species, and these low-diversity communities are generally associated with optimal health outcomes (8–13). Conversely, diversity is very common in the gut and is considered a hallmark of gut health (14, 15). Several prevalent vaginal bacteria have only recently been characterized or remain uncharacterized (e.g., bacterial vaginosis associated bacterium 1 [BVAB1; proposed name "*Candidatus* Lachnocurva vaginae"], BVAB2, *Gardnerella* spp., *Mageeibacillus indolicus*, and *Megasphaera lornae*) (16–19). Finally, reference sequence databases contain data from >10-fold-more gut samples than vaginal samples (20). Given these differences in community composition and structure and that the vaginal microbiome is less well characterized than the gut microbiome, it is unclear how well metagenome inference methods might perform for the vaginal microbiome. It is also unclear whether these methods may introduce biases that are specific to analyses of the vaginal microbiome. To fill these gaps, we compared the performance of PICRUSt2 and Tax4Fun2 using paired 16S rRNA gene amplicon sequencing and whole-metagenome sequencing (WMGS) data from vaginal samples from pregnant individuals. We examined whether performance varied across sample characteristics (hierarchical cluster) and gene characteristics (functional category) to describe the bias PICRUSt2- and Tax4Fun2-inferred metagenomes may introduce into analyses.

## RESULTS

**Description of participants, microbiota clusters, and metagenomes.** We used data from the Pregnancy, Infection, and Nutrition (PIN) cohort to evaluate and compare the performance of PICRUSt2 and Tax4Fun2 (2, 3, 21). PIN cohort participants were recruited from prenatal clinics in North Carolina, USA, between 1995 and 2008. Participants were assigned female sex at birth and were eligible if they were ≥16 years old and at ≤29 weeks of gestation with a singleton pregnancy. Seventy-two participants were selected for this analysis from 3,063 PIN participants with live births, recorded delivery dates, and stored vaginal swabs: 35 cases who experienced early preterm birth (PTB) (<32 weeks of gestation at delivery), and 37 controls who experienced term birth (37 to 41 weeks of gestation at delivery) (see Fig. S1 in the supplemental material). Participant demographics, gestational age at sample collection, and Nugent score category at sample collection did not differ substantially between PTB cases and term birth controls (Table 1).

Vaginal swabs were collected between 24 and 29 weeks of gestation, and stored swabs were used for 16S rRNA gene amplicon sequencing targeting the V1-to-V3

**TABLE 1** PIN participant characteristics overall and by birth outcome

| Characteristic | Result for participants[a] | | | | | |
| | Overall (n = 72) | | Preterm birth cases (n = 35) | | Term birth controls (n = 37) | |
| | n | % | n | % | n | % |
|---|---|---|---|---|---|---|
| Self-reported race | | | | | | |
| Black | 45 | 63 | 22 | 63 | 23 | 62 |
| White | 27 | 38 | 13 | 37 | 14 | 38 |
| Age (yr)[b] | 26 | 21–30 | 26 | 21–29 | 26 | 22–33 |
| Gestational age (wk) at[b]: | | | | | | |
| Sample collection | 27 | 26–28 | 27 | 25–28 | 28 | 26–28 |
| Delivery | 37 | 30–40 | 30 | 28–31 | 40 | 38–41 |
| Nugent score at sample collection[c] | | | | | | |
| Non-BV (0–3) | 50 | 69 | 25 | 71 | 25 | 68 |
| Intermediate (4–6) | 14 | 19 | 7 | 20 | 7 | 19 |
| BV (7–10) | 8 | 11 | 3 | 9 | 5 | 14 |
| Hierarchical cluster[d] | | | | | | |
| L. crispatus dominated | 17 | 24 | 11 | 31 | 6 | 16 |
| L. iners dominated | 31 | 43 | 13 | 37 | 18 | 49 |
| Mixed | 24 | 33 | 11 | 31 | 13 | 35 |

[a]Preterm birth is <32 weeks of gestation, and term birth is between 37 and 41 weeks of gestation.
[b]Continuous characteristics are presented as median (n) and interquartile range (%).
[c]BV, bacterial vaginosis.
[d]Based on hierarchical clustering of 16S rRNA gene amplicon sequencing data.

hypervariable regions and WMGS. The average 16S rRNA gene amplicon sequencing depth was 31,049 reads/sample (range, 804 to 54,628), and 115 amplicon sequence variants (ASVs) were assigned. In hierarchical clustering of 16S rRNA gene amplicon sequencing data using Jensen-Shannon divergence distances, we identified three clusters of vaginal microbiota composition based on *a priori* knowledge of common vaginal bacterial community types and visual inspection of the clustering dendrogram and microbiota composition stacked bar plot (Fig. 1). According to two of three internal validation statistics (connectivity and silhouette), the optimal number of clusters was two. The third internal validation statistic (Dunn) suggested the optimal number of clusters was three. With two clusters, *Lactobacillus iners*-dominated samples and mixed-composition samples belonged to the same cluster. Because *L. iners* dominance is commonly observed (8–12), we decided to differentiate between *L. iners*-dominated samples and mixed samples and use three clusters instead of two. Seventeen samples (24% of total) belonged to a cluster dominated by *Lactobacillus crispatus*, including 11 PTB cases (31% of cases). Thirty-one (43% of total) belonged to a cluster dominated by *L. iners*, including 13 PTB cases (37% of cases). Twenty-four (33% of total) belonged to a cluster of samples with mixed composition, including 11 PTB cases (31% of cases). According to all $\alpha$-diversity metrics estimated, samples from the *Lactobacillus*-dominated clusters appeared to have similar distributions of $\alpha$ diversity (Fig. 2). The $\alpha$-diversity values for samples in the mixed cluster overlapped with those from both *Lactobacillus*-dominated clusters, although the mixed cluster showed the widest ranges.

The average WMGS depth was 1,029,551 total reads/sample (range, 311,519 to 6,226,056) with 71,628 mapped reads/sample (range, 424 to 823,811) (150-bp reads), and 2,031 KEGG (Kyoto Encyclopedia of Genes and Genomes) orthologs (KOs) were assigned. At the highest level of KO functional category, observed metagenomes of samples belonging to the *L. crispatus*-dominated cluster appeared to be the most homogeneous, and they appeared to be more similar to metagenomes of samples belonging to the mixed cluster than to metagenomes of samples belonging to the *L. iners*-dominated

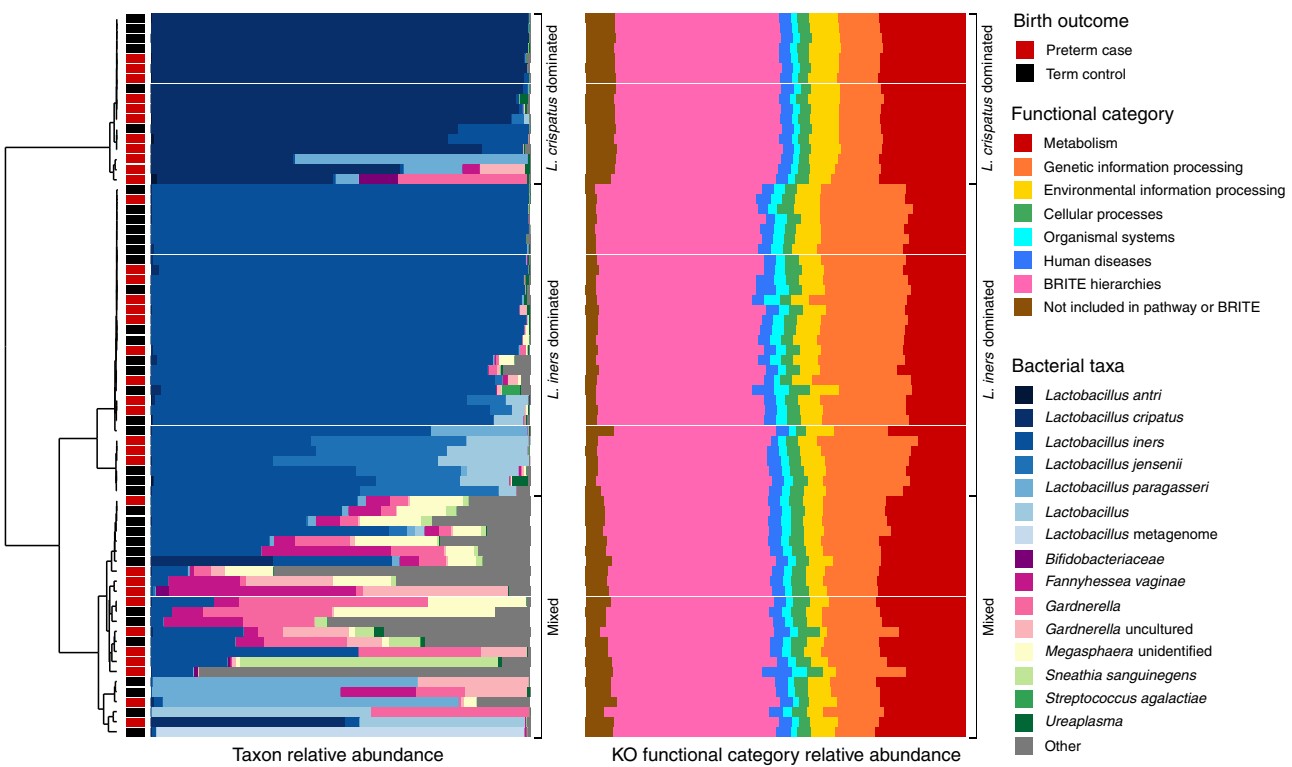

**FIG 1** Microbiota and metagenome composition and birth outcome of 72 PIN participants included in the analysis. Shown is a dendrogram from hierarchical clustering of 16S rRNA gene amplicon sequencing data; the resulting clusters are indicated by the brackets to the right of each stacked bar plot. The taxon relative abundance from 16S rRNA gene amplicon sequencing data and the KO functional category relative abundance from observed whole-metagenome sequencing data are presented.

cluster (Fig. 1; Fig. S2 has a stacked bar plot colored by the second-highest level of KO functional category). Compared to the other clusters, *L. iners*-dominated metagenomes appeared to be enriched with genes involved in genetic information processing and had lower relative abundances of genes involved in metabolism and uncharacterized genes. $\alpha$ diversity and clusters estimated from WMGS KO relative abundances are summarized and compared to estimates from 16S rRNA gene amplicon sequencing data in Text S1, Fig. S4-S7, and Table S2.

We inferred metagenome content from 16S rRNA gene amplicon sequencing data using PICRUSt2 and Tax4Fun2 according to the developers' recommendations (2, 3). PICRUSt2 predicted the presence of 5,882 KOs, of which 1,503 (26%) were observed in WMGS data (Fig. 3). Tax4Fun2 predicted the presence of 7,049 KOs, of which 1,506 (21%) were observed. A total of 1,490 KOs were observed in WMGS data and predicted by PICRUSt2 and Tax4Fun2, and all evaluations and comparisons of metagenome inference performance were restricted to these 1,490 KOs.

**Metagenome inference performance varied by dominant *Lactobacillus* species.** We evaluated metagenome inference performance using two approaches. First, we estimated KO-specific Spearman correlation coefficients between observed and predicted KO relative abundances. Second, we estimated Spearman correlation coefficients between univariable hypothesis test *P* values estimated using observed and predicted KO relative abundances. This approach was proposed by Sun et al. following their observation that Spearman correlation coefficients between observed and predicted KO relative abundances are insensitive to randomly permutating the data, indicating they may be an unreliable measure of metagenome inference performance (6). We used Wilcoxon tests to test the null hypothesis that KO relative abundances do not differ between PTB cases and term birth controls. We performed Wilcoxon tests using observed and predicted KO relative

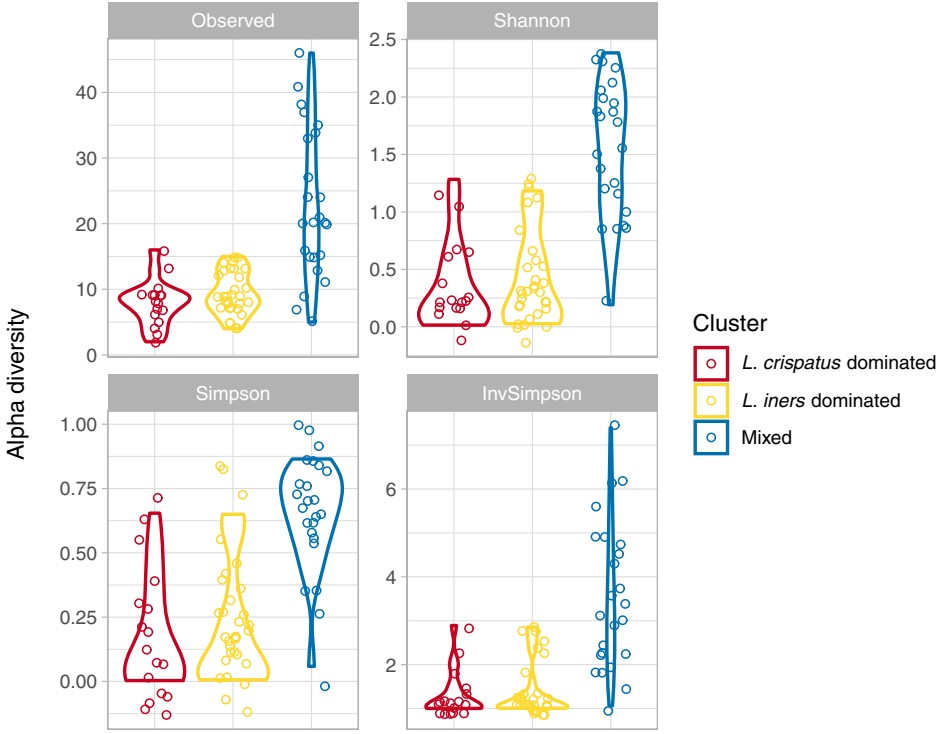

**FIG 2** Vaginal microbiota $\alpha$ diversity. $\alpha$ diversity was estimated using 16S rRNA gene amplicon sequencing data. The $\alpha$-diversity metric is indicated by the label at the top of each panel. "Observed" refers to the number of ASVs observed in a sample or the observed richness. Results were stratified and colored by hierarchical cluster from hierarchical clustering of 16S rRNA gene amplicon sequencing data.

abundances separately, and we transformed *P* values according to the following equation to capture the significance and direction of KO relative abundance differences:

$$P_t = \log_{10}(P) \times \text{sign}\left[\left(\overline{\text{KO}} \mid \text{PTB}\right) - \left(\overline{\text{KO}} \mid \text{term birth}\right)\right]$$

In this equation, $P_t$ is the transformed *P* value, *P* is the Wilcoxon test *P* value, $\overline{\text{KO}} \mid$ PTB is the mean KO relative abundance among PTB cases, and $\overline{\text{KO}} \mid$ term birth is the mean KO relative abundance among term birth controls. We performed all analyses with the original observed and predicted metagenome data, as well as with 100 random permutations of the observed and predicted metagenome data in which KO relative abundances were independently permuted across samples, which serves as a robustness check (6).

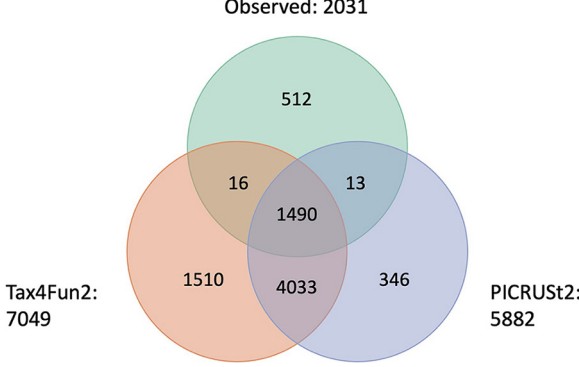

**FIG 3** Number and overlap of KOs observed and predicted. "Observed" refers to KOs annotated by HUMAnN2 in the whole-metagenome sequencing data.

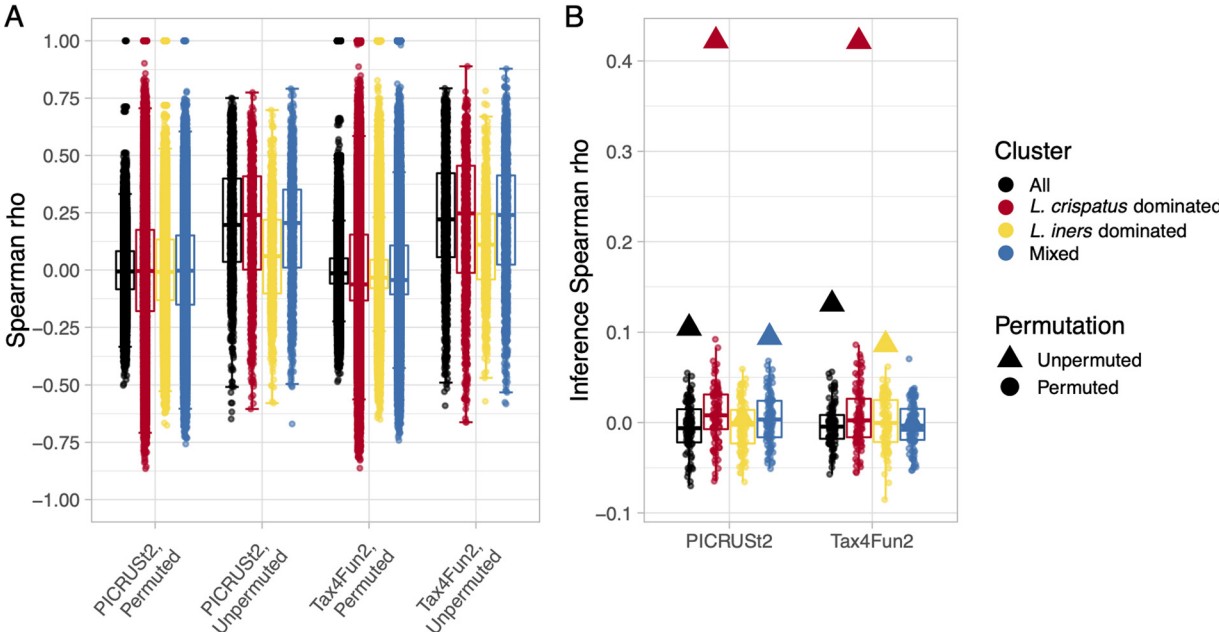

**FIG 4** Vaginal microbiome metagenome inference performance varies across hierarchical cluster. Shown are clusters from hierarchical clustering of 16S rRNA gene amplicon sequencing data. Analyses were performed with the original observed and predicted metagenome data and with 100 random permutations of observed and predicted metagenome data (KO relative abundances independently permuted across samples), which serves as a robustness check. Spearman correlation coefficients between observed and predicted KO relative abundances, stratified by hierarchical cluster, metagenome inference method, and permutation, are shown in panel A. Spearman correlation coefficients between univariable hypothesis test transformed *P* values estimated with observed and predicted KO relative abundances, stratified by hierarchical cluster, metagenome inference method, and permutation, are shown in panel B. We used Wilcoxon tests to test the null hypothesis that KO relative abundances do not differ between PTB cases and term birth controls. We performed Wilcoxon tests using observed and predicted KO relative abundances separately, and we transformed *P* values according to the equation from the section "Metagenome inference performance varied by dominant *Lactobacillus* species" to capture the significance and direction of KO relative abundance differences.

The median Spearman correlation between observed and PICRUSt2-predicted KO relative abundances was 0.20 (range, −0.65 to 0.75) (Fig. 4A). We stratified by hierarchical cluster to evaluate whether PICRUSt2 or Tax4Fun2 performance varied across common vaginal microbiota community types. The median correlation was similar among the *L. crispatus*-dominated (0.24) and mixed (0.21) clusters and lower among the *L. iners*-dominated cluster (0.06). We observed a similar pattern for the Tax4Fun2 predictions, although median correlations were higher (overall, 0.22 [range, −0.59 to 0.79]; *L. crispatus*-dominated, 0.25; mixed, 0.24; *L. iners*-dominated, 0.11). Correlations were not robust to permutation (median correlations of ∼0 for permuted data), indicating this may be a reliable method for evaluating metagenome inference performance for the vaginal microbiome.

For the hypothesis test comparing KO relative abundances between cases and controls, transformed *P* value correlations were highest among *L. crispatus*-dominated samples (PICRUSt2, 0.42; Tax4Fun2, 0.42) (Fig. 4B). Correlations were low overall (PICRUSt2, 0.10; Tax4Fun2, 0.13) among *L. iners*-dominated samples (PICRUSt2, 0.00; Tax4Fun2, 0.09) and among mixed samples (PICRUSt2, 0.09; Tax4Fun2, 0.00). Correlations were not robust to permutation, and variation across clusters did not appear to be influenced by transformed *P* value outliers (Fig. S3). Taken together, these data indicate that PICRUSt2 and Tax4Fun2 perform best among *L. crispatus*-dominated microbiotas and poorly among *L. iners*-dominated microbiotas.

PICRUSt2 provides a weighted nearest-sequenced-taxon index (wNSTI), a sample-specific measure of relatedness between observed ASVs and their nearest neighbors in the PICRUSt2 reference database, weighted by sample composition (3). The median wNSTI overall was 0.16, and while medians were similar among *L. crispatus*- and *L. iners*-dominated samples (0.16 and 0.20, respectively), *L. iners*-dominated samples showed much wider range of values and substantial density of samples at values greater than the maximum wNSTI for the

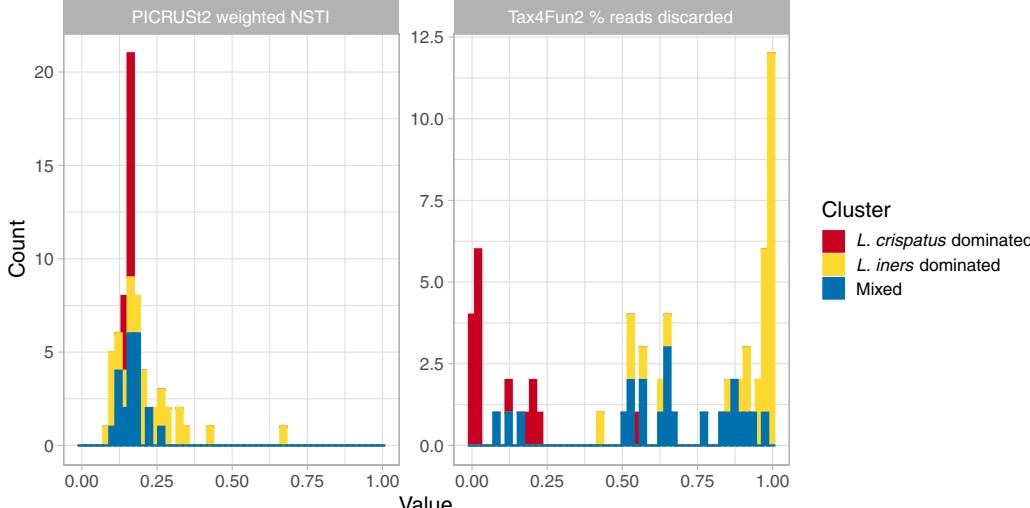

**FIG 5** ASVs observed in *L. iners*-dominated samples are less well represented in metagenome inference reference databases. Shown are clusters from hierarchical clustering of 16S rRNA gene amplicon sequencing data. The weighted NSTI (nearest-sequenced-taxon index) was estimated by PICRUSt2 as a sample-specific measure of relatedness between observed ASVs and their nearest neighbors in the PICRUSt2 reference database, weighted by sample composition. Lower values indicate an ASV is more closely related to its nearest neighbor. In Tax4Fun2, ASVs that are unrepresented in the reference database and that have <97% ANI to their nearest reference neighbor's 16S rRNA gene are discarded prior to metagenome inference.

*L. crispatus* cluster (0.17) (Fig. 5). In Tax4Fun2, ASVs that are unrepresented in the reference database and that have <97% average nucleotide identity (ANI) to their nearest neighbor are discarded prior to metagenome inference (2). The median proportion of discarded reads was 66% overall, with 3% discarded among *L. crispatus*-dominated samples and 97% discarded among *L. iners*-dominated samples (Fig. 5).

**Metagenome inference performance varied across KO function.** We also stratified by the highest level of KO functional category to evaluate whether PICRUSt2 or Tax4Fun2 performance varied across groups of related genes. Median Spearman correlations between observed and PICRUSt2-predicted KO relative abundances were similar across KO functional categories (range from 0.13 for genetic information processing to 0.20 for BRITE hierarchies and uncharacterized) (Fig. 6A). We also observed little variation in correlations for Tax4Fun2 predictions, though correlations were slightly higher (range from 0.16 for organismal systems to 0.25 for genetic information processing). Correlations were not robust to permutation.

We observed more variation across KO functional categories in correlations between transformed *P* values estimated with observed and predicted KO relative abundances (Fig. 6B). For the hypothesis test comparing KO relative abundances between cases and controls, correlations were highest for uncharacterized KOs and metabolism (PICRUSt2, 0.18 and 0.16, respectively; Tax4Fun2, 0.22 and 0.14, respectively). PICRUSt2 performed worst for genetic information processing (−0.08), and Tax4Fun2 performed worst for organismal systems (−0.25). Correlations were not robust to permutation, and variation across KO functional categories did not appear to be influenced by transformed *P* value outliers (Fig. S3). These data indicate that PICRUSt2 and Tax4Fun2 may perform best for uncharacterized genes and poorly for genes involved in genetic information processing.

***L. crispatus*-dominated metagenomes were enriched with good-performance KOs, and *L. iners*-dominated metagenomes were enriched with poor-performance KOs.** We examined differences in observed metagenome content between *L. crispatus*- and *L. iners*-dominated samples as a potential cause of differential metagenome inference performance between these clusters. *L. crispatus* relative abundance was strongly positively correlated with uncharacterized KO relative abundance (linear model, $R^2$ = 50%), whereas *L. iners* relative abundance was strongly negatively correlated with uncharacterized KO relative abundance (linear model, $R^2$ = 75%) (Fig. 7). *L. crispatus* relative abundance was

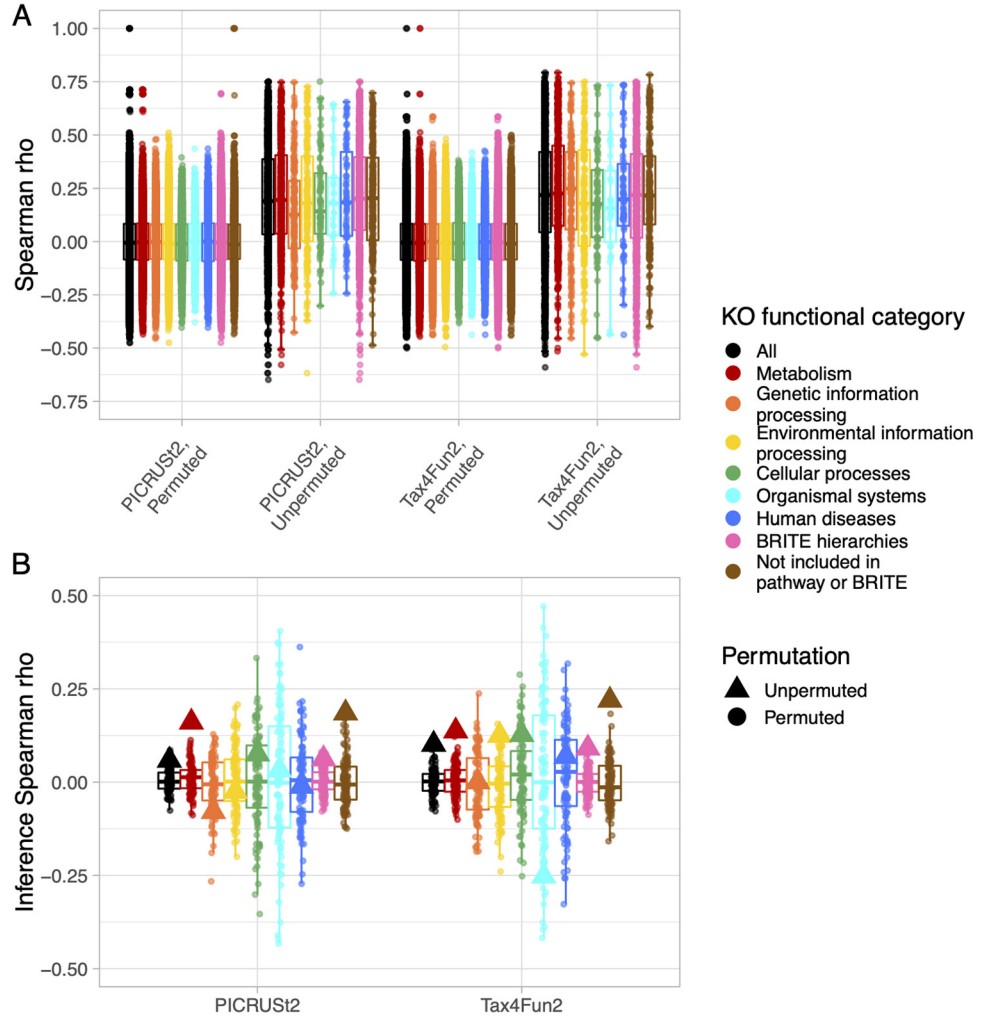

**FIG 6** Vaginal microbiome metagenome inference performance varies across KO functional category. Correlation analyses were performed with original observed and predicted metagenome data and with 100 random permutations of observed and predicted metagenome data (KO relative abundances independently permuted across samples), which serves as a robustness check. Spearman correlation coefficients between observed and predicted KO relative abundances, stratified by KO functional category, metagenome inference method, and permutation, are shown in panel A. Spearman correlation coefficients between univariable hypothesis test transformed *P* values estimated with observed and predicted KO relative abundances, stratified by KO functional category, metagenome inference method, and permutation, are shown in panel B. We used Wilcoxon tests to test the null hypothesis that KO relative abundances do not differ between PTB cases and term birth controls. We performed Wilcoxon tests using observed and predicted KO relative abundances separately, and we transformed *P* values according to the equation from the section "Metagenome inference performance varied by dominant *Lactobacillus* species" in order to capture the significance and direction of KO relative abundance differences.

strongly negatively correlated with genetic information processing KO relative abundance (linear model, $R^2 = 45\%$), while *L. iners* relative abundance was strongly positively correlated with genetic information processing KO relative abundance (linear model, $R^2 = 75\%$) (Fig. 7).

## DISCUSSION

While the performance of various metagenome inference methods have been compared in several human body site and environmental ecosystems, to our knowledge this is the first study specific to the vaginal microbiome (1–7). Our cluster-stratified analysis demonstrated that PICRUSt2 and Tax4Fun2 perform best among *L. crispatus*-dominated communities and poorly among *L. iners*-dominated communities. We attribute this differential performance to biological differences between the communities, as well as reference differences between the communities. In terms of biological difference, our KO functional category-stratified analysis highlighted that *L. iners*-dominated microbiotas tend to be

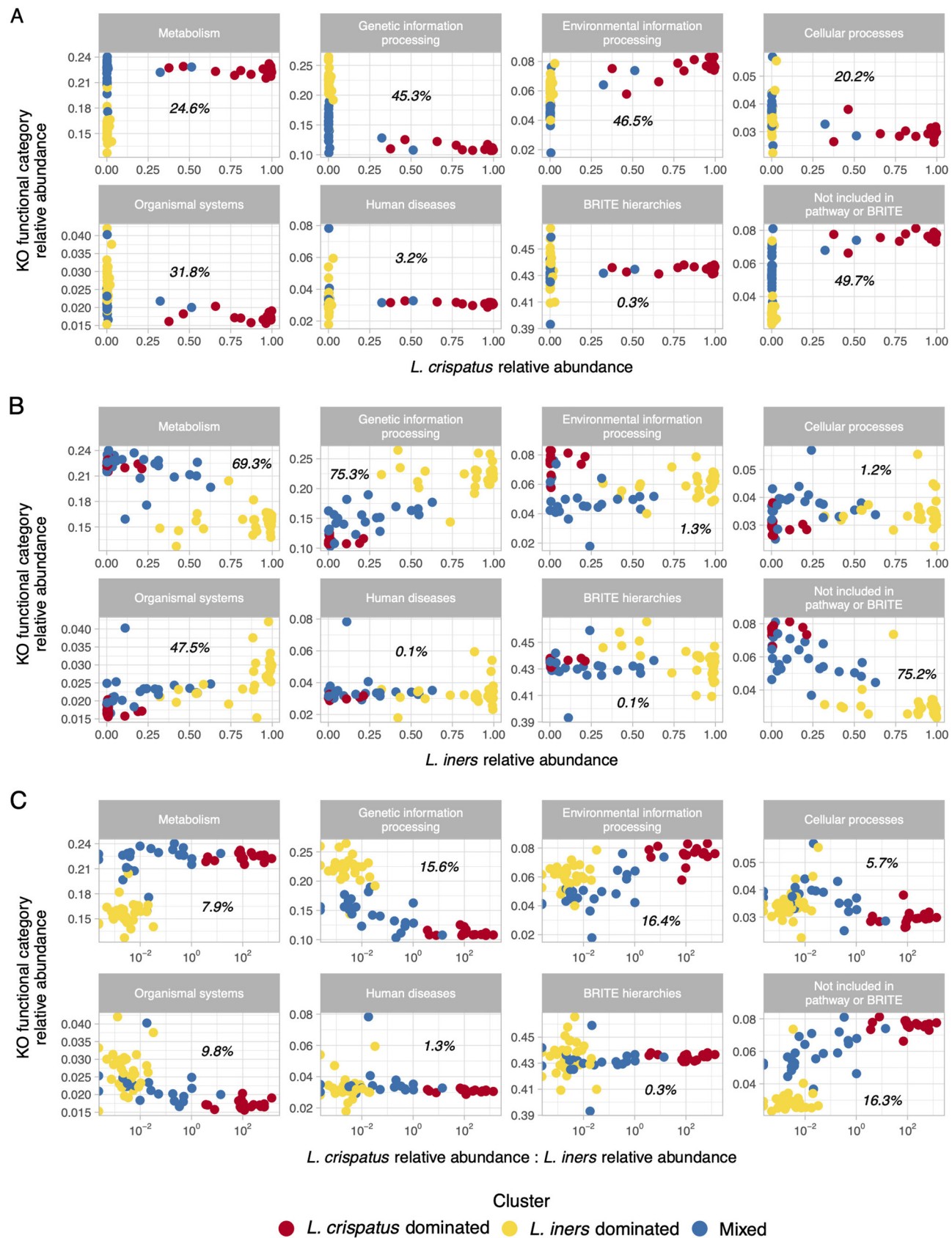

**FIG 7** Correlations between KO functional category relative abundances and *L. iners* and *L. crispatus* relative abundances. Scatterplots of *L. crispatus* relative abundance versus KO functional category relative abundance with one panel for each KO functional category are shown in panel A, *L. iners*

enriched with KO functional categories that are inferred poorly, namely, genetic information processing. This is consistent with a prior genomic comparison of 15 *L. iners* and 15 *L. crispatus* strains. The *L. iners* core and accessory genomes contained larger proportions of genes involved in replication and repair, transcription, and translation than the *L. crispatus* core and accessory genomes (22). Within their respective clusters in the current analysis, *L. iners* and *L. crispatus* dominated communities with averages of 86% and 87% relative abundance, respectively. It follows that differences in genome composition between the species will yield differences in metagenome composition between *L. iners*- and *L. crispatus*-dominated communities (in general and with respect to genetic information processing genes). This appears to be a driver of poor metagenome inference performance among *L. iners*-dominated communities. It is unclear why PICRUSt2 and Tax4Fun2 performed poorly for genetic information processing genes. One prior function-stratified analysis of gut microbiome samples found that the original implementation of PICRUSt, PICRUSt2, and the original implementation of Tax4Fun tended to perform well for genetic information processing genes, with the exception of genes involved in transcription (6).

A second biological difference is related to interstrain variation. Marker gene sequencing is largely incapable of resolving taxa to the strain/subspecies level. Instead, heterogeneous strains are collapsed into a single higher-order feature, and interstrain variation is structurally removed from the data. If we cannot identify strains in 16S rRNA gene amplicon sequencing data, we cannot infer strain-specific genome content, and interstrain differences in gene content are structurally removed from inferred metagenome data. *L. iners* is estimated to have more interstrain variation than other vaginal lactobacilli (22–25), so this limitation will more seriously impact gene content inference for *L. iners* and metagenome inference for *L. iners*-dominated communities than it will for *L. crispatus*.

In terms of reference differences between *L. iners*- and *L. crispatus*-dominated communities, *L. iners* is less well represented than *L. crispatus* in the reference databases used in the current analysis. HUMAnN2 was used to characterize gene families in the observed WMGS data, and the HUMAnN2 protein reference database was generated from UniRef90 and UniRef50 (prior to 2017, it was unclear which release) (26). As of 31 December 2016, UniRef90 contained 13,078 *L. crispatus* proteins and 3,205 *L. iners* proteins, and UniRef50 contained 6,146 *L. crispatus* proteins and 1,677 *L. iners* proteins. *L. crispatus* genomes are ~1.5- to 2-fold larger and contain ~1.5- to 2-fold more predicted coding sequences than *L. iners* genomes (22, 24); however, there are 4.1-fold more and 3.7-fold more *L. crispatus* proteins in UniRef90 and UniRef50, respectively. *L. iners* is substantially underrepresented in UniRef90/UniRef50, so it is similarly underrepresented in the HUMAnN2 protein reference database. KO sequences uniquely originating from *L. iners* are less likely to be identified in our data than those uniquely originating from *L. crispatus*. However, this bias does not directly impact our correlation analysis results. Because this bias results in underdetection of *L. iners* KOs in the observed data, there are no "gold standard" data for these KOs, and no comparison can be made with inferred metagenome data if the KOs are predicted to be present.

The PICRUSt2 reference database was constructed using bacterial genomes in the Integrated Microbial Genomes & Microbiomes database on 8 November 2017 (3), which contained 15 *L. crispatus* genomes and 17 *L. iners* genomes. Despite similar representation, our observed *L. crispatus* ASV was more closely related to its nearest reference neighbor than our observed *L. iners* ASV was to its nearest reference neighbor (Fig. 4). (Neither ASV was distantly related enough to be discarded prior to metagenome inference). Unlike

**FIG 7** Legend (Continued)

relative abundance versus KO functional category relative abundance with one panel for each KO functional category is shown in panel B, and the ratio of *L. crispatus* relative abundance to *L. iners* relative abundance versus KO functional category relative abundance with one panel for each KO functional category is shown in panel C. The italicized text in each panel presents the $R^2$ value from a linear model regressing KO functional category relative abundance (response variable) on *L. crispatus* relative abundance (A), *L. iners* relative abundance (B), or the ratio of *L. crispatus* relative abundance to *L. iners* relative abundance (C) (explanatory variables). The $R^2$ values represent the proportion of variation in KO relative abundance explained by the variation in *L. crispatus* relative abundance (A), *L. iners* relative abundance (B), or the ratio of *L. crispatus* relative abundance to *L. iners* relative abundance (C). Points are colored by hierarchical cluster from hierarchical clustering of 16S rRNA gene amplicon sequencing data.

for HUMAnN2, bias introduced by the PICRUSt2 reference database differentially impacts *L. crispatus*- and *L. iners*-dominated communities, contributing to poorer performance for *L. iners*-dominated communities. The Tax4Fun2 reference database was constructed using all complete genomes in NCBI RefSeq on 18 August 2018 (2), which contained 6 *L. crispatus* genomes and 0 *L. iners* genomes. Tax4Fun2 identifies each observed ASV's nearest neighbor in its reference database and infers gene content for ASVs that have ≥97% sequence identity to their nearest neighbor; the remaining ASVs and associated reads are discarded (2). Without reference data for *L. iners*, >41% of reads were discarded from each *L. iners*-dominated sample prior to metagenome inference, and 75% of *L. iners*-dominated samples had ≥90% of reads discarded (Fig. 4). Among *L. crispatus*-dominated samples, most had <25% of reads discarded, and at most 55% of reads were discarded. Clearly, the *L. iners* ASV we observed is not sufficiently closely related to its nearest reference neighbor for it to contribute to Tax4Fun2 metagenome inference. Despite massive loss of reads from *L. iners*-dominated samples in Tax4Fun2 and no loss in PICRUSt2, Tax4Fun2 performed somewhat better for *L. iners*-dominated communities, which likely reflects differences in accuracy of genome content inference for other taxa present.

Taking an epidemiologic lens to these results, we can consider differential metagenome inference performance to be analogous to differential measurement error, resulting in differential misclassification. Differential measurement error occurs when the degree or direction of outcome (response variable) measurement error differs across exposure (explanatory variable) statuses, or when the degree or direction of exposure measurement error differs across outcome statuses. Differential measurement error and subsequent differential misclassification bias effect estimates, and this bias is hard to predict in that it can be in either direction (toward or away from the null value) (27, 28). On the other hand, nondifferential measurement error and nondifferential misclassification (independent of exposure or outcome status) typically bias effect estimates toward the null (27, 28). To qualify as differential measurement error, metagenome inference performance must differ across exposure or outcome status or across a factor that is associated with exposure or outcome status. While cluster or community type will not be the exposure or outcome of interest in every vaginal microbiome study that uses metagenome inference, we posit that cluster will be associated with the exposure or the outcome in virtually all of these studies based on the large body of evidence supporting such associations (9, 10, 12, 29–31). It follows that metagenome inference will introduce differential measurement error, differential misclassification, and hard-to-predict bias in vaginal microbiome research. Results from these analyses should be interpreted with substantial caution and the understanding that they may over- or underestimate associations with metagenome content. This is particularly concerning given the high global prevalence of *L. iners*-dominated vaginal microbiotas (8–12); poor metagenome inference among these communities will impact similarly large proportions of study populations.

Several studies have compared the performance of various metagenome inference methods using gut microbiome samples, and we can use these reports to benchmark PICRUSt2 and Tax4Fun2 performance for the vaginal microbiome. We will consider results based on observed and predicted KO relative abundance correlations because few studies have examined *P* value correlations (5, 6), and assessing performance based on *P* value correlations depends on the hypothesis tested and the true effect size. Among stool samples from various populations, four analyses of six metagenome inference methods report median correlations of ~0.60 to 0.90 (3–6). In our analysis, median relative abundance correlations were 0.19 and 0.22 (Fig. 4A and Fig. 5A), indicating substantially worse performance for the vaginal microbiome than for the gut microbiome. Poorer metagenome inference for the vaginal microbiome can be attributed, likely in large part, to the vaginal microbiome being relatively understudied compared to the gut microbiome. A recent investigation of all human microbiome data sets in the Sequence Read Archive, DNA Data Bank of Japan, and European Nucleotide Archive revealed that 220,017 (49%) of 444,829 samples were from the gut, compared to 17,784 (4%) from the vagina (20). Given its prevalence (8–12), underrepresentation of

*L. iners* in metagenome inference reference databases contributes to poorer performance for the vaginal microbiome than the gut. Additionally, a taxon formerly considered one species, *Gardnerella vaginalis*, was recently identified to the species level as four named species (*Gardnerella leopoldii*, *Gardnerella piotii*, *Gardnerella swidsinskii*, and *G. vaginalis*) and 9 genomic species (19). *Gardnerella* species cannot be differentiated by 16S rRNA gene sequences, so the genomic diversity of *Gardnerella* species cannot be captured by metagenome inference. Considering the relative abundance of *Gardnerella* spp. was as high as 50% in this analysis (an average of 18% within the mixed cluster), inability to accurately capture *Gardnerella* species' genome content will decrease metagenome inference performance.

Our results also indicate poorer metagenome inference performance for the vaginal microbiome than has been reported for the urogenital microbiome. One study examined the first implementations of PICRUSt and Tax4Fun using urogenital samples and reported median relative abundance correlation of ~0.65 (4). Reasons for this discrepancy are unclear because additional details on the urogenital samples were not reported.

Our analysis has several strengths. This is the first published metagenome inference comparison specific to the vaginal microbiome. Focusing on the vaginal microbiome enabled us to identify patterns of metagenome inference performance that directly relate to vaginal microbial ecology, as well as important implications for risk of bias in vaginal microbiome research that employs metagenome inference. We were able to confidently identify this pattern because we consistently observed differential metagenome inference performance across clusters in both of the evaluations we performed (relative abundance and *P* value correlations). Using both evaluation methods is a second strength of this analysis. Finally, 16S rRNA gene amplicon sequencing and WMGS depths achieved in the parent study indicate the high quality of the sequencing data used.

Our findings should also be interpreted in the context of the study's limitations. With 72 participants, we identified only three clusters of vaginal microbiota composition. With a larger population, we may have been able to characterize metagenome inference performance for more vaginal microbiota community types observed in U.S. populations (e.g., *Lactobacillus gasseri* dominated, *Lactobacillus jensenii* dominated, subgroups of *Lactobacillus* dominance, and diverse bacterial vaginosis [BV]-like communities) (30). Second, all participants were pregnant and enrolled in North Carolina, USA. This limits generalizability of our results to nonpregnant populations and populations in other regions, especially considering documented differences in vaginal microbiota composition by pregnancy status (32–35) and between global populations (36–40). Finally, this analysis was restricted to metagenome features that were observed and predicted. Totals of 541 (27%) observed KOs, 4,392 (75%) PICRUSt2-predicted KOs, and 5,559 (79%) Tax4Fun2-predicted KOs did not contribute to the analysis (Fig. 3).

Our analysis demonstrates that metagenome inference methods perform more poorly for the vaginal microbiome than for the gut microbiome. Performance was differential according to vaginal microbiota cluster, with the best performance among *L. crispatus*-dominated communities and the worst among *L. iners*-dominated communities. Genome content differences between *L. iners* and *L. crispatus* and underrepresentation of *L. iners* in reference databases appear to drive differential performance, which will result in differential measurement error, differential misclassification, and hard-to-predict bias in vaginal microbiome research that employs metagenome inference. As the cost of whole-metagenome sequencing continues to fall, investigators will ideally be able to characterize the actual vaginal bacterial metagenome, obviating the need for and eliminating measurement error introduced by metagenome inference.

## MATERIALS AND METHODS

We used paired 16S rRNA gene amplicon sequencing and WMGS data from the PIN cohort to evaluate and compare the performance of PICRUSt2 and Tax4Fun2 for the vaginal microbiome (2, 3, 21). The PIN cohort is a prospective cohort study approved by the University of North Carolina Institutional Review Board (protocol 16-2166, with continuous approval since August 2016), and all participants provided written informed consent prior to enrollment. For this secondary analysis, we selected participants from among those who consented to additional testing of stored specimens. We report our results according to the Strengthening The Organization and Reporting of Microbiome Studies (STORMS) guidelines (Table S1) (41).

**PIN participants and study procedures.** PIN cohort participants were recruited from prenatal clinics at the Wake County Human Services Department, WakeMed Medical Center/Wake Area Health Education Center, and University of North Carolina hospitals in North Carolina between 1995 and 2008. Potential participants were eligible if they were assigned female sex at birth, ≥16 years old, and at ≤29 weeks of gestation with a singleton pregnancy. Potential participants were ineligible if they did not plan to continue care or deliver at the study site, did not have telephone access, or were non-English speaking. Study staff identified potential participants through medical record review and approached potential participants about enrolling.

Participants were enrolled at ≤29 weeks of gestation and followed until delivery. Demographic characteristics were recorded at enrollment, and clinicians collected two Dacron-tipped swabs from the posterior vaginal apex between 24 and 29 weeks of gestation. One swab was used for evaluation of BV according to Nugent score and then placed in a Digene Virapap tube with transport medium (Digene Diagnostics, Inc., Silver Spring, MD) (42). The second swab was placed in a Roche Amplicor specimen collection tube containing extraction buffer (Roche Diagnostic Systems, Inc., Branchburg, NJ). Specimens were refrigerated prior to transport and frozen at −70°C within 6 h of collection. Following enrollment, a telephone questionnaire was used to collect information on reproductive and medical history—sexual behaviors during pregnancy and tobacco, alcohol, and other drug use. Gestational age at delivery was defined by early ultrasound (completed <20 weeks of gestation) for 90% of participants and by last menstrual period date for the remaining participants. Birth outcomes were abstracted from medical records.

Participants were selected for this substudy in a nested case-control design. Cases were participants who experienced early preterm birth at <32 weeks of gestation, and controls were randomly selected from among participants who experienced term birth between 37 and 41 weeks of gestation. Cases and controls were selected from among participants with adequate extracted DNA for WMGS and ≥800 high-quality reads from 16S rRNA gene amplicon sequencing.

**Microbiome data generation.** Stored vaginal swabs were used for 16S rRNA gene amplicon sequencing as previously reported (43–45). Frozen swabs were thawed on ice, DNA was extracted using the PowerSoil DNA isolation kit (Qiagen, Hilden, Germany) according to manufacturer recommendations, and extracted DNA was quantified using PicoGreen. Extracted DNA was amplified with barcoded primers targeting the 16S rRNA gene V1-to-V3 hypervariable regions using protocols established by the Virginia Commonwealth University Vaginal Human Microbiome Project (46). The primers contain an Illumina linkage adaptor, unique barcode (8 bases for forward primer, 12 for reverse), a variable sequence spacer (0 to 6 bases), and 16S rRNA gene primers (full primer sequences are available in reference 47). The forward primer was a 4:1 mixture of primers Fwd-P1 and Fwd-P2. The reverse primer was Rev1B (46, 47). Amplicon samples were multiplexed (384 samples/run) using a sample-specific dual-index strategy and sequenced on the Illumina MiSeq platform (2 × 300 base-paired-end protocol) (Illumina, San Diego, CA). Resulting amplicon sequence data were processed using QIIME2 (version 2019.1.0) and the DADA2 denoise-single method, truncating reads at 120 bases (48, 49). Taxonomy was assigned using the Ribosomal Database Project Naïve Bayesian Classifier and SILVA database (release 138.1) (50–52).

Extracted DNA was also used for whole-metagenome library preparation. Paired-end metagenomic DNA libraries were prepared from 250 ng of genomic DNA using the Accel-NGS 2S Plus DNA library kit (Integrated DNA Technologies, Coralville, IA) with an insert size of ~350 bp. Samples were preprocessed as previously described (44, 45). Barcoded libraries were multiplexed (10 to 11 samples/lane) and sequenced on the Illumina HiSeq 4000 (2 × 150-base-paired-end reads). Raw sequences were demultiplexed using Illumina's bcl2fastq software and quality filtered using MEEPTOOLS (53, 54), where reads shorter than 70 bases and with a MEEP quality score of >1 were excluded. WMGS data were screened for duplicate reads, and human sequences were removed by aligning reads to the hg19 build of the human genome using the BWA aligner (55). The remaining reads were processed and aligned according to the approach recommended by Martin and colleagues (56). Reads with <60 nonmasked, non-low-complexity bases were considered low complexity and discarded (56). The remaining reads were aligned using the CLC Assembly Cell aligner (Qiagen, Hilden, Germany) requiring 80% identity over 75% of the query length (56). Gene families and functional pathways were characterized using HUMAnN2 and the KEGG Orthology database (26, 57–59).

**Statistical analysis.** We used descriptive statistics to summarize participant characteristics, including maternal age, self-reported race, and gestational ages at vaginal swab collection and delivery. We estimated vaginal microbiota $\alpha$ diversity based on 16S rRNA gene amplicon sequencing data using the estimate_richness function of the phyloseq package (v.1.34.0 throughout) in R (v.4.0.4 throughout). We categorized vaginal microbiota composition by hierarchical clustering of 16S rRNA gene amplicon sequencing data based on Jensen-Shannon divergence distances and Ward linkage using the distance function of the phyloseq package, hclust and cutree functions of the stats package (v.4.0.4), and dendsort function of the dendsort package (v.0.3.4) in R. We used the same functions to estimate $\alpha$ diversity and identify clusters based on WMGS KO relative abundances, and we compared $\alpha$ diversity and cluster membership between 16S rRNA gene amplicon sequencing and WMGS data.

We inferred metagenome content from 16S rRNA gene amplicon sequencing data using PICRUSt2 and Tax4Fun2 according to the developers' recommendations/default parameters (2, 3). We selected PICRUSt2 and Tax4Fun2 because their original implementations, PICRUSt and Tax4Fun, are highly cited among metagenome inference methods (as of 15 February 2023, PICRUSt and Tax4Fun original publications were cited 7,309 and 1,119 times, respectively) (1, 4). We selected their second implementations, PICRUSt2 and Tax4Fun2, because they demonstrated marked performance improvements over the original implementations (1–4). We implemented PICRUSt2 (v.2.4.1) in Miniconda (v.4.9.2). In addition to inferred metagenome composition, PICRUSt2 computes wNSTI, a sample-specific measure of relatedness between observed ASVs and their nearest neighbors in the PICRUSt2 reference database, weighted by

sample composition (3). We implemented Tax4Fun2 with the default reference database and 99% clustering threshold using the runRefBlast and makeFunctionalPrediction functions of the Tax4Fun2 package (v.1.1.5) in R. For ASVs unrepresented in the Tax4Fun2 reference database, the nearest neighbor in the database is identified, and the ASVs are assumed to have the same gene content as their nearest neighbor when the 16S rRNA gene ANI is ≥97% (2). ASVs with a <97% ANI are discarded prior to metagenome inference (2). Tax4Fun2 provides the proportion of ASVs and reads discarded.

For metagenome inference performance evaluation and comparison, we restricted the observed, PICRUSt2-inferred, and Tax4Fun2-inferred metagenome data sets to KOs present at relative abundance of >0% in all three data sets. We evaluated metagenome inference performance using two approaches. First, we estimated KO-specific Spearman correlation coefficients between observed and predicted KO relative abundances. Second, we estimated Spearman correlation coefficients between univariable hypothesis test $P$ values estimated using observed and predicted KO relative abundances. This approach was proposed by Sun et al. following their observation that Spearman correlation coefficients between observed and predicted KO relative abundances are insensitive to randomly permutating the data, indicating they may be an unreliable measure of metagenome inference performance (6). We used Wilcoxon tests to test the null hypothesis that KO relative abundances do not differ between PTB cases and term birth controls. We performed Wilcoxon tests using observed and predicted KO relative abundances separately, and we transformed $P$ values according to the equation shown above in the section "Metagenome inference performance varied by dominant *Lactobacillus* species" to capture the significance and direction of KO relative abundance differences.

For both evaluation approaches, we performed two stratified analyses to evaluate whether metagenome inference performance differs according to vaginal microbiota hierarchical cluster (from 16S rRNA gene amplicon sequencing) or KO functional category (highest-level categories). We performed each analysis with the original observed and predicted metagenome data, as well as with 100 random permutations of the observed and predicted metagenome data in which KO relative abundances were independently permuted across samples, which serves as a robustness check (6). We used linear model $R^2$ values to examine relationships between KO functional category relative abundances (response variables) and *L. crispatus* relative abundances, *L. iners* relative abundances, and the ratio of *L. crispatus* to *L. iners* relative abundances (explanatory variables). We examined PICRUSt2 wNSTI values and the proportion of reads discarded in Tax4Fun2 as additional measures of sample-specific performance.

**Data availability.** The 16S and shotgun metagenome sequencing data can be accessed at the NCBI Sequence Read Archive (SRA) under BioProject no. PRJNA876771 (https://www.ncbi.nlm.nih.gov/bioproject/PRJNA876771/). The R script, R data files, and corresponding codebook/data dictionary used in the analysis are available on GitHub (https://github.com/kaycart/mg-inference-comp).

## SUPPLEMENTAL MATERIAL

Supplemental material is available online only.

**TEXT S1**, PDF file, 0.01 MB.
**FIG S1**, PDF file, 0.1 MB.
**FIG S2**, PDF file, 0.04 MB.
**FIG S3**, PDF file, 0.8 MB.
**FIG S4**, PDF file, 0.02 MB.
**FIG S5**, PDF file, 0.04 MB.
**FIG S6**, PDF file, 0.1 MB.
**FIG S7**, PDF file, 0.1 MB.
**TABLE S1**, PDF file, 0.2 MB.
**TABLE S2**, PDF file, 0.01 MB.

## ACKNOWLEDGMENTS

We thank the participants and staff of the Pregnancy, Infection, and Nutrition cohort.

The Pregnancy, Infection, and Nutrition cohort was supported by grants from NIH/NICHD (HD37584, HD39373) and NIH/NIDDK (DK61981). The 16S rRNA gene amplicon sequencing study (nested within the Pregnancy, Infection, and Nutrition cohort) was funded in part by grants from NIH/NIMHD (R01MD011504) and NIH/NIEHS (P30 ES010126). The General Clinic Research Center was supported by National Institutes of Health General Clinical Research Centers Program of the Division of Research Resources grant RR00046. K.A.C. receives support through an STD/AIDS Research Training Predoctoral Fellowship (NIH T32 AI07140). The funders had no role in study design, data collection and interpretation, or the decision to submit this work for publication.

J.E.B. has received an honorarium from Ferring Pharmaceuticals. G.A.B. is on the Scientific Advisory Board of Juno Bio, Ltd. All other authors declare no conflict of interest.

S.M.E. contributed to the Pregnancy, Infection, and Nutrition (PIN) cohort design, implementation, and data collection. G.A.B., J.M.F., and M.G.S. performed 16S rRNA gene

amplicon sequencing and whole-metagenome sequencing. G.A.B., J.M.F., M.G.S., S.S., and A.Z. contributed to 16S gene amplicon sequencing data processing and whole-metagenome sequencing data processing. A.A.F., J.E.B., K.A.C., M.C.W., and S.S. contributed to the metagenome inference comparison conceptualization. A.A.F., K.A.C., M.C.W., and S.S. contributed to statistical analysis. K.A.C. wrote the first manuscript draft and led manuscript writing. All authors contributed to, reviewed, and approved the final manuscript.

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
