## [Reviewer comments · mSystems]

Vaginal microbiome metagenome inference accuracy: differential measurement error according to community composition

Kayla Carter, Anthony Fodor, Jennifer Balkus, Angela Zhang, Myrna Serrano, Gregory Buck, Stephanie Engel, Michael Wu, and Shan Sun

Corresponding Author(s): Kayla Carter, University of Maryland Baltimore

Review Timeline:

Submission Date:	October 21, 2022
Editorial Decision:	December 23, 2022
Revision Received:	February 17, 2023
Accepted:	February 21, 2023

Editor: Jack Gilbert

Reviewer(s): Disclosure of reviewer identity is with reference to reviewer comments included in decision letter(s). The following individuals involved in review of your submission have agreed to reveal their identity: Kara Wiggin (Reviewer #2)

Transaction Report:

DOI: <https://doi.org/10.1128/msystems.01003-22>

December 23, 2022

Dr. Kayla A Carter
University of Maryland, Baltimore
Institute for Genome Sciences
670 W Baltimore St
Room 3060
Baltimore, MD 21201

Re: mSystems01003-22 (Vaginal microbiome metagenome inference accuracy: differential measurement error according to community composition)

Dear Dr. Kayla A Carter:

Thank you for submitting your manuscript to mSystems. We have completed our review and I am pleased to inform you that, in principle, we expect to accept it for publication in mSystems. However, acceptance will not be final until you have adequately addressed the reviewer comments.

Preparing Revision Guidelines

Sincerely,

Jack Gilbert

Editor, mSystems

Journals Department
American Society for Microbiology
1752 N St., NW

Reviewer comments:

Reviewer #1 (Comments for the Author):

The study by Carter and co-authors describe an interesting measurement error according to community composition when functional information is inferred from 16S rRNA gene amplicon data of vaginal samples. The paper is well written and structured. However, a couple of missing aspects should be addressed within a minor revision of the manuscript before acceptance.

Major concerns:

First of all the authors state (and reference literature) which shows the superior performance to infer functional information of the microbiome from targeted approaches. It would be great if the authors compare their own performance for vaginal samples to other biomes - give values and also use the same methods, since the authors highlighted their methodological setup as one of their strengths within this study. How much worse is metagenomics inference for vaginal samples compared to samples from stool or soil etc.?

Since *L. crispatus* and *L. iners* dominated vaginal microbiotas were shown to be responsible for differential performance of metagenomics inference, the authors should investigate how well these key taxa are represented in HUMAnN2 (for shotgun metagenomics), PICRUSt2 and Tax4Fun2 (for amplicon data). Which and how many reference genomes of *L. crispatus* and *L. iners* are covered in those databases? Any knowledge or estimates available of strain diversity of those two *Lactobacilli* species? Could strain level diversity impact metagenomics inference as well? Please elaborate on these questions in your results and discussion sections.

Regarding the small sample size - wouldn't it be possible to ease exclusion criteria and test the validity of the observations on a larger sample size as well?

Minor concerns:

Page 6, line 124: V1-V3 hypervariable regions is too vague - please provide the name, position and sequence of the used primer construct and a reference - either here or at least in the method section (see page 20, line 450).

Page 6, line 132: revise - ...optimal number of clusters was set to two...

Page 8, line 171: I'm skeptical if simple Spearman correlations would be the best method of choice here. I would recommend to use methods which are aware of the sparse compositional nature of microbiome datasets. Could you test with sparCC for instance as well?

Page 12, line 264: correct to - involved in replication and repair

Page 13, line 293 to page 14, line 319:

This part is interesting but way too detailed, I would recommend to shorten it move parts to the M&M section or to the Supplementary instead.

Page 21, line 466: Use "demultiplexed" instead. I would avoid the term "binned" here as this is usually used for the reconstruction of genomes from metagenomics data.

Page 21, line 472: Provide further details of the quality filtering settings - a reference is not enough.

Page 24, line 537: Please provide all the details on your data and scripts with this GitHub repo when submitting a revised version of your manuscript.

Table 1: The % header of the table is misleading as in other columns the values refer to the interquartile range. I recommend to use a reference to inform if the value refers to % or something else. In addition, it would be helpful to test if differences between preterm birth cases and term birth controls were significant for the reported values and scores.

Figure 3: I would recommend to use the name of the annotation tool (HUMAnN2) instead of "observed" as this term suggest to show "true" values and not just values of another method.

Reviewer #2 (Comments for the Author):

Overall, this paper is a well written and well analyzed study on the accuracy of metagenome inference tools such as PICRUSt2 and Tax4Fun2. The authors results highlighting the issues with these tools for vaginal microbiome samples are well thought out and presented and emphasizes the importance of considering vaginal microbiome community type when conducting these analyses.

I appreciated the author's inclusion of multiple versions of the spearman correlation analysis to demonstrate the effectiveness of the chosen method for vaginal microbiome samples, and thought the analysis overall was well done.

In the statistical analysis section of the methods section (line 476-486) the author notes that they used the 16S data to calculate alpha diversity and construct the vaginal microbiome community types. I would have liked to see the comparison of these results between the 16S data and the WMGS data. I am curious whether the author saw any differences in alpha diversity between these two data sets, and particularly in the delineation of the vaginal microbiome community types. As WMGS is more sensitive I would be curious if more or different community types would be identified and whether this would change any of the conclusions of the paper. From the taxa barplots I do appreciate and understand the community types that were chosen but would have liked to see this comparison mentioned in the paper, or at least laid out in the supplemental material, perhaps as justification for the choice to use 16S data for these analyses. I think that would make this section even stronger.

In Figure 4 and 5 I think your figure caption would benefit from clarification of your permutation legend section for those who are not as familiar with this method.

Finally, I appreciate that the author will be uploading R scripts and R data files to GitHub! I cannot see this file yet so I will just stress the importance of this, as well as in some way providing the raw results of the spearman analyses, particularly the univariate hypothesis test section. If this will not be included in the GitHub I encourage the author to include this in the supplemental section.

Reviewer #1 (Comments for the Author):

The study by Carter and co-authors describe an interesting measurement error according to community composition when functional information is inferred from 16S rRNA gene amplicon data of vaginal samples.

The paper is well written and structured. However, a couple of missing aspects should be addressed within a minor revision of the manuscript before acceptance.

Major concerns:

1. First of all the authors state (and reference literature) which shows the superior performance to infer functional information of the microbiome from targeted approaches. It would be great if the authors compare their own performance for vaginal samples to other biomes - give values and also use the same methods, since the authors highlighted their methodological setup as one of their strengths within this study. How much worse is metagenomics inference for vaginal samples compared to samples from stool or soil etc.?

We agree that this would be an interesting comparison; however, additional analyses of other body site/environmental microbiomes is beyond the scope of the manuscript. Our primary interest was to describe how well these methods perform specifically for the vaginal microbiome because this had not previously been reported and there was good reason to believe the methods might perform worse for the vaginal microbiome. Additionally, as our results demonstrate that these methods generally perform poorly for the vaginal microbiome and will introduce bias in hypothesis testing, we feel that quantifying the differential in performance between the vaginal microbiome and other body site/environmental microbiomes is not of particular interest. Most investigators considering using one of these methods are more likely to consider whether to use a metagenome inference method or which method to use, not whether they should use the method with vaginal microbiota 16S data or other body site/environmental microbiota 16S data.

2. Since *L. crispatus* and *L. iners* dominated vaginal microbiotas were shown to be responsible for differential performance of metagenomics inference, the authors should investigate how well these key taxa are represented in HUMAnN2 (for shotgun metagenomics), PICRUSt2 and Tax4Fun2 (for amplicon data). Which and how many reference genomes of *L. crispatus* and *L. iners* are covered in those databases? Any knowledge or estimates available of strain diversity of those two Lactobacilli species? Could strain level diversity impact metagenomics inference as well? Please elaborate on these questions in your results and discussion sections.

Thank you for raising this point. Below, we provide a detailed response which is summarized in the revised discussion (lines 338-398) and Figure 5 (new).

We used HUMAnN2 to characterize gene families and functional pathways (not to assign taxonomy). The HUMAnN2 protein reference database was based on UniRef90 and UniRef50. The date that these data were accessed is not provided in the HUMAnN2 publication or its supplemental materials. The supplement does describe an analysis conducted with “novel genomes” that were deposited in 2017, after constructing the reference database, so sequences were accessed at some point during/before 2017. By December 31, 2016, UniRef90 contained 13,078 *L. crispatus* proteins, 3,205 *L. iners* proteins, and (as an additional point of reference) 13,199 *Gardnerella* spp. proteins. By

December 31, 2016, UniRef50 contained 6,146 *L. crispatus* proteins, 1,677 *L. iners* proteins, and 5,566 *Gardnerella* spp. proteins. Each of these values were the same when considering a December 31, 2017 cutoff. *L. crispatus* and *Gardnerella* genomes are approximately 1.5-2x larger than and contain approximately 1.3-2x more predicted coding sequences than *L. iners* genomes. However, there are 4.1x more *L. crispatus* and *Gardnerella* proteins in UniRef90, and 3.3-3.7x more in UniRef50. *L. iners* is clearly and substantially underrepresented in UniRef90 and UniRef50, so it is similarly underrepresented in the HUMAnN2 protein reference database. As such, KO sequences uniquely originating from *L. iners* are less likely to be identified in our data than KO sequences uniquely originating from *L. crispatus* or *Gardnerella*.

The PICRUSt2 reference database is based on bacterial and archaeal genomes in the IMG database on November 8, 2017. On this date, IMG contained 15 *L. crispatus* genomes, 17 *L. iners* genomes, and 41 *Gardnerella* spp. genomes. Here, *L. crispatus* and *L. iners* are similarly represented. PICRUSt2 infers genome content by first placing each ASV within a pre-computed phylogenetic tree and identifying its nearest neighbors. Gene content is estimated for ASVs that are not represented in the tree or reference database as a weighted average of gene content for the ASV's nearest neighbors. PICRUSt2 estimates a nearest-sequenced taxon index (NSTI) as a measure of relatedness between unrepresented ASVs and their nearest neighbor, and ASVs with values >2 are discarded (this is the default setting the authors note this is a lenient filter intended to remove “problematic sequences,” we observed no NSTI >2 so no ASVs or reads were discarded from metagenome inference). The following table and histogram summarize NSTI overall and according to cluster (histogram is new Figure 5). The difference in distribution of NSTI between *L. crispatus*- and *L. iners*-dominated samples is striking. While medians are similar (0.16 vs. 0.18), *L. iners*-dominated samples showed much wider range of values and substantial density of samples at values greater than the maximum wNSTI for the *L. crispatus* cluster (0.172). Because *L. crispatus* and *L. iners* dominated samples in their respective clusters with a mean of 86% and 87% relative abundance, it is reasonable to assume that these two species are the main drivers of NSTI values in these clusters. These NSTI data suggest that while *L. iners* and *L. crispatus* are similarly represented in the PICRUSt2 reference database, our observed *L. iners* ASVs are more distantly related to their nearest neighbors than our *L. crispatus* ASVs. It is possible that the genomes included in the reference database are more representative of the *L. crispatus* core genome or a “typical” *L. crispatus* genome than they are for *L. iners*. (We added details on NSTI to the methods, lines 743-745, 810-811, and results, lines 241-247).

	NSTI	
	Median	Interquartile range
Overall	0.162	0.143 – 0.197
L. crispatus -dominated	0.160	0.154 – 0.161
L. iners -dominated	0.180	0.137 – 0.262
Mixed	0.165	0.134 – 0.183

The Tax4Fun2 reference database is based on all complete genomes and genomes with status “chromosome” in NCBI RefSeq on August 18, 2018. On this date, RefSeq contained 6 *L. crispatus* genomes, 0 *L. iners* genomes, and 12 *Gardnerella* spp. genomes. The Tax4Fun2 reference database does not contain any gene content information for *L. iners*. Tax4Fun2 identifies each ASV’s nearest neighbor in the reference database and infers gene content for ASVs that have $\geq 97\%$ sequence identity with their nearest neighbor. Remaining ASVs and associated reads are discarded. The histogram above presents the proportion of reads discarded from each sample, and the table below summarizes the proportion of ASVs and reads discarded. Focusing on discarded reads, *L. crispatus*-dominated communities have the lowest proportion of discarded reads (maximum 0.55). However, 75% of *L. iners*-dominated communities had 90% of reads discarded, and all *L. iners*-dominated samples had $>41\%$ of reads discarded. The proportion of reads discarded among mixed samples is intermediate between *L. crispatus*- and *L. iners*-dominated samples. These data clearly demonstrate that the *L. iners* ASV we observed is not sufficiently closely related to its nearest neighbor in the reference database for it to contribute to metagenome inference. (We added detail on the proportion of reads discarded to the methods, lines 748-752, 810-811, and results, 247-251).

	Tax4Fun2 % discarded ASVs		Tax4Fun2 % discarded reads	
	Median	IQR	Median	IQR
Overall	0.64	0.49 – 0.75	0.66	0.21 – 0.97
L. crispatus -dominated	0.33	0.26 – 0.44	0.03	0.01 – 0.19
L. iners -dominated	0.72	0.57 – 0.79	0.97	0.90 – 0.99
Mixed	0.70	0.62 – 0.74	0.66	0.54 – 0.86

The underrepresentation of *L. iners* compared to *L. crispatus* in the HUMAnN2 protein reference database and Tax4Fun2 reference database, and insufficient sequence identity between our *L. iners* ASV and the PICRUSt2 reference database nearest neighbor likely all contribute to poorer metagenome inference performance among *L. iners*-dominated samples. However, despite massive loss of reads from *L. iners*-dominated samples in Tax4Fun2 and no loss of reads in PICRUSt2, Tax4Fun2 performed better than PICRUSt2

for *L. iners* dominated communities (and generally across the board), which is likely related to differences in how metagenome content is inferred.

Strain-level diversity certainly impacts metagenome inference accuracy. Because 16S amplicon sequencing is largely unable to differentiate strains within a species, strains within a species are grouped into a single higher-order feature, and inter-strain variation is structurally removed from the data. If we cannot identify strains in 16S data, we cannot infer strain-specific genome content, and inter-strain differences in gene content are likewise structurally removed from the inferred data. *L. iners* is estimated to have more inter-strain variation than other vaginal lactobacilli, so this limitation will more seriously impact *L. iners* than *L. crispatus* and likely also contributes to poorer metagenome inference accuracy among *L. iners*-dominated samples.

3. Regarding the small sample size - wouldn't it be possible to ease exclusion criteria and test the validity of the observations on a larger sample size as well?

Unfortunately, we are not able to increase the sample size by easing eligibility criteria. Whole metagenome sequencing was only performed for the subset of PIN participants selected for the nested case control study of associations between the vaginal bacterial metagenome and preterm birth. We do not have resources for metagenome sequencing of additional samples.

Minor concerns:

4. Page 6, line 124: V1-V3 hypervariable regions is too vague - please provide the name, position and sequence of the used primer construct and a reference - either here or at least in the method section (see page 20, line 450).

We added this information to the materials and methods section (lines 668-672).

5. Page 6, line 132: revise - ...optimal number of clusters was set to two...

Lines 138-140 summarize the internal validation statistics from hierarchical clustering. We then explain our decision to use three clusters (lines 140-144) based on disagreement between internal validation statistics and meaningfully different microbiota compositions being lumped into a single cluster when only two clusters were used. We feel this is communicated clearly and chose not to revise this text.

6. Page 8, line 171: I'm skeptical if simple Spearman correlations would be the best method of choice here. I would recommend to use methods which are aware of the sparse compositional nature of microbiome datasets. Could you test with sparCC for instance as well?

Thank you for raising this point. In our analysis, we are interested in KO-specific/within-KO correlations comparing a given KO's observed and inferred relative abundances across samples. Because we are not investigating correlations between different KOs, a method like sparCC that accounts for the sparse compositional nature of WMGS data when estimating correlations between features is not necessary. Additionally, prior published

metagenome inference comparisons use Spearman correlation coefficients to evaluate performance, and using the same method in our analysis promotes comparison of our results to prior reports.

7. Page 12, line 264: correct to - involved in replication and repair

Thank you for catching this error, we corrected it.

8. Page 13, line 293 to page 14, line 319: This part is interesting but way too detailed, I would recommend to shorten it move parts to the M&M section or to the Supplementary instead.

We respectfully disagree with the reviewer. We feel that the previously undescribed measurement error that we observed (and subsequent bias in estimating associations and hypothesis testing using inferred metagenome data) is a major finding of our manuscript. We chose to provide ample detail in describing the issue at hand because mSystems is not primarily an epidemiology/public health journal, and much of its readership may not be familiar with these concepts as they apply specifically to estimating associations and/or hypothesis testing. In addition to being a major finding, we also feel this text is best placed in the discussion as opposed to materials and methods or supplement because we are not describing measurement error and bias in our metagenome inference comparison analysis, but instead in inferential analyses that rely on inferred metagenome data. We chose not to trim this text or move it to another section.

9. Page 21, line 466: Use "demultiplexed" instead. I would avoid the term "binned" here as this is usually used for the reconstruction of genomes from metagenomics data.

Thank you, we revised as suggested (line 685).

10. Page 21, line 472: Provide further details of the quality filtering settings - a reference is not enough.

We added details on filtering of low-complexity reads and parameters used in aligning metagenomic reads (lines 708-711).

11. Page 24, line 537: Please provide all the details on your data and scripts with this GitHub repo when submitting a revised version of your manuscript.

All code and data files used in this analysis as well as a codebook/data dictionary are on GitHub (<https://github.com/kaycart/mq-inference-comp>), and we added this link to the Data availability section of the manuscript.

12. Table 1: The % header of the table is misleading as in other columns the values refer to the interquartile range. I recommend to use a reference to inform if the value refers to % or something else. In addition, it would be helpful to test if differences between preterm birth cases and term birth controls were significant for the reported values and scores.

In the first submission, we included a footnote indicating that continuous data were presented as median and interquartile range. Originally, this footnote was only indexed at each row containing continuous data, we revised the table to additionally index this footnote with the % headers. In the current manuscript, we are not specifically interested in drawing inference regarding differences between PIN participants who experienced preterm vs. term births, we only use these comparisons to evaluate metagenome inference performance. As such, we decided not to include results from hypothesis tests comparing characteristics between cases and controls. A recent publication (Sun et al mSystems 2022 <https://doi.org/10.1128/msystems.00017-22>) thoroughly investigates these differences in the PIN cohort, and it is cited in our current manuscript (reference 43).

13. Figure 3: I would recommend to use the name of the annotation tool (HUMAnN2) instead of "observed" as this term suggest to show "true" values and not just values of another method.

Thank you for this suggestion. We feel the term “observed” is appropriate to refer to the data generated by whole metagenome sequencing, and throughout the manuscript, we make the distinction between observed data and PICRUSt2/Tax4Fun2-inferred data. For these reasons, we chose not to revise the figure or the text. We instead added a comment to the Figure 3 legend indicating that “Observed” refers to data KOs annotated by HUMAnN2 in the whole metagenome sequencing data.

Reviewer #2 (Comments for the Author):

Overall, this paper is a well written and well analyzed study on the accuracy of metagenome inference tools such as PICRUSt2 and Tax4Fun2. The authors results highlighting the issues with these tools for vaginal microbiome samples are well thought out and presented and emphasizes the importance of considering vaginal microbiome community type when conducting these analyses.

I appreciated the author's inclusion of multiple versions of the spearman correlation analysis to demonstrate the effectiveness of the chosen method for vaginal microbiome samples, and thought the analysis overall was well done.

1. In the statistical analysis section of the methods section (line 476-486) the author notes that they used the 16S data to calculate alpha diversity and construct the vaginal microbiome community types. I would have liked to see the comparison of these results between the 16S data and the WMGS data. I am curious whether the author saw any differences in alpha diversity between these two data sets, and particularly in the delineation of the vaginal microbiome community types. As WMGS is more sensitive I would be curious if more or different community types would be identified and whether this would change any of the conclusions of the paper. From the taxa barplots I do appreciate and understand the community types that were chosen but would have liked to see this comparison mentioned in the paper, or at least laid out in the supplemental material, perhaps as justification for the choice to use 16S data for these analyses. I think that would make this section even stronger.

Thank you for raising this point. Below, we provide a detailed response which is reported in the revised methods (lines 171-173) and supplemental results (new).

Yes, in working through our analysis we also estimated alpha diversity and clustered samples based on KO relative abundances from WMGS sequencing. We identified two clusters of samples based on KO relative abundances. Unlike in the 16S analysis, all three internal validation statistics agreed that two clusters were optimal, and also unlike 16S clusters, we have no established paradigm of commonly observed vaginal WMGS clusters, so we elected to use two clusters for WMGS data. The first cluster included 42 samples (21 PTB cases) and was marked by relative enrichment (compared to the second cluster) of KOs involved in metabolism and uncharacterized KOs. The second cluster included 30 samples (14 PTB cases) and was marked by relative enrichment of genetic information processing genes. The following two plots are the equivalents of Figure 1 and Supplemental Figure 2 from the manuscript, but displaying the dendrogram from WMGS hierarchical clustering, ordered according to the dendrogram from WMGS hierarchical clustering, and indicating the WMGS clusters (these are the new Supplemental Figures 4-5).

The figure below shows that, while ranges of KO alpha diversity values overlapped between the WMGS clusters, samples belonging to the metabolism/uncharacterized-KO-enriched cluster (metagenome cluster 1, pink) tended to have higher alpha diversity than

samples belonging to the genetic-information-processing-enriched cluster (metagenome cluster 2, orange) (new Supplemental Figure 6).

The table below cross-tabulates 16S and WMGS clusters (new Supplemental Table 2). The metabolism/ uncharacterized-KO-enriched cluster included all *L. crispatus*-dominated samples, all but one mixed sample, and two *L. iners*-dominated samples. The genetic-information-processing-enriched cluster included the remaining *L. iners*-dominated and mixed samples.

	Metabolism, uncharacterized KO enrichment (N = 42)		Genetic information processing KO enrichment (N = 30)	
	n	%	n	%
L. crispatus -dominated	17	40	0	0
L. iners -dominated	2	5	29	97
Mixed	23	55	1	3

The figure above shows that, within 16S clusters, KO alpha diversity was largely similar/overlapping between *L. crispatus*-dominated and mixed samples, which tended to show higher alpha diversity than *L. iners*-dominated samples. *L. crispatus*-dominated samples showed the narrowest range and most right-shifted distribution (highest values) of KO alpha diversity, which is in contrast with alpha diversity estimated from 16S sequencing, both in our data and as has been observed in the field more broadly.

The following scatter plots depict the relationship between alpha diversity estimated from 16S data (x axis) and WMGS data (y axis) (new Supplemental Figure 7). On the whole, there appear to be null-to-positive relationships between 16S and KO alpha diversity, depending on the diversity metric considered. Within clusters (16S and WMGS), no clear or consistent trends emerge, with several trend lines being nonmonotonic or approximately flat.

A

B

Considering these data, we elected not to run an additional analysis comparing metagenome inference performance stratified by WMGS cluster. Because WMGS cluster 1 was almost exclusively comprised of *L. crispatus*-dominated and mixed samples while WMGS cluster 2 was almost exclusively comprised of *L. iners*-dominated samples, and because differential metagenome inference performance in our analysis was driven by differences in metagenome content between *L. crispatus*- and *L. iners*-dominated samples, we expect PICRUSt2 and Tax4Fun2 to perform better for WMGS cluster 1 than for WMGS cluster 2. Because we feel this is a reasonable assumption and because WMGS clusters

are less easily interpretable/clinically relevant than the 16S clusters, we feel an additional analysis stratified by WMGS cluster would not add substantial value to the manuscript.

2. In Figure 4 and 5 I think your figure caption would benefit from clarification of your permutation legend section for those who are not as familiar with this method.

Thank you for this suggestion. In both captions, we added a comment describing how data were permuted and that these permuted data were used as a robustness check.

3. Finally, I appreciate that the author will be uploading R scripts and R data files to GitHub! I cannot see this file yet so I will just stress the importance of this, as well as in some way providing the raw results of the spearman analyses, particularly the univariate hypothesis test section. If this will not be included in the GitHub I encourage the author to include this in the supplemental section.

All code and data files used in this analysis as well as a codebook/data dictionary are on GitHub (<https://github.com/kaycart/mq-inference-comp>), and we added this link to the Data availability section of the manuscript. Hypothesis testing and running correlations are included in the code, so their results are not directly provided in the code files.

February 21, 2023

Dr. Kayla A Carter
University of Maryland Baltimore
Institute for Genome Sciences
670 W Baltimore St
Room 3060
Baltimore, MD 21201

Re: mSystems01003-22R1 (Vaginal microbiome metagenome inference accuracy: differential measurement error according to community composition)

Dear Dr. Carter:

Your manuscript has been accepted, and I am forwarding it to the ASM Journals Department for publication. For your reference, ASM Journals' address is given below. Before it can be scheduled for publication, your manuscript will be checked by the mSystems production staff to make sure that all elements meet the technical requirements for publication. They will contact you if anything needs to be revised before copyediting and production can begin. Otherwise, you will be notified when your proofs are ready to be viewed.

If you would like to submit a potential Featured Image, please email a file and a short legend to msystems@asmusa.org. Please note that we can only consider images that (i) the authors created or own and (ii) have not been previously published. By submitting, you agree that the image can be used under the same terms as the published article. File requirements: square dimensions (4" x 4"), 300 dpi resolution, RGB colorspace, TIF file format.

We recognize that the video files can become quite large, and so to avoid quality loss ASM suggests sending the video file via <https://www.wetransfer.com/>. When you have a final version of the video and the still ready to share, please send it to mSystems staff at msystems@asmusa.org.

Sincerely,

Jack Gilbert
Editor, mSystems

Journals Department
E-mail: mSystems@asmusa.org